# A Customized Efficient Deep Learning Model for the Diagnosis of Acute Leukemia Cells Based on Lymphocyte and Monocyte Images

**Sanam Ansari [1], Ahmad Habibizad Navin [2,\*], Amin Babazadeh Sangar [1] [ID], Jalil Vaez Gharamaleki [3] and Sebelan Danishvar [4] [ID]**

[1] Department of Computer Engineering, Urmia Branch, Islamic Azad University, Urmia 5716963896, Iran
[2] Department of Computer Engineering, Tabriz Branch, Islamic Azad University, Tabriz 5157944533, Iran
[3] Hematology and Oncology Research Center, Tabriz University of Medical Science, Tabriz 5165687386, Iran
[4] College of Engineering, Design and Physical Sciences, Brunel University London, Uxbridge UB8 3PH, UK
[\*] Correspondence: a.habibzad@srbiau.ac.ir

**Abstract:** The production of blood cells is affected by leukemia, a type of bone marrow cancer or blood cancer. Deoxyribonucleic acid (DNA) is related to immature cells, particularly white cells, and is damaged in various ways in this disease. When a radiologist is involved in diagnosing acute leukemia cells, the diagnosis is time consuming and needs to provide better accuracy. For this purpose, many types of research have been conducted for the automatic diagnosis of acute leukemia. However, these studies have low detection speed and accuracy. Machine learning and artificial intelligence techniques are now playing an essential role in medical sciences, particularly in detecting and classifying leukemic cells. These methods assist doctors in detecting diseases earlier, reducing their workload and the possibility of errors. This research aims to design a deep learning model with a customized architecture for detecting acute leukemia using images of lymphocytes and monocytes. This study presents a novel dataset containing images of Acute Lymphoblastic Leukemia (ALL) and Acute Myeloid Leukemia (AML). The new dataset has been created with the assistance of various experts to help the scientific community in its efforts to incorporate machine learning techniques into medical research. Increasing the scale of the dataset is achieved with a Generative Adversarial Network (GAN). The proposed CNN model based on the Tversky loss function includes six convolution layers, four dense layers, and a Softmax activation function for the classification of acute leukemia images. The proposed model achieved a 99% accuracy rate in diagnosing acute leukemia types, including ALL and AML. Compared to previous research, the proposed network provides a promising performance in terms of speed and accuracy; and based on the results, the proposed model can be used to assist doctors and specialists in practical applications.

**Keywords:** blood cancer; leukemia; white blood cells; deep learning

## 1. Introduction

Blood comprises a viscous substance called plasma and floating cells produced by the bone marrow. Bone marrow is a spongy, soft substance found within bones. It contains stem cells, which are in charge of making blood cells. Blood cells are divided into three types: white blood cells, which guard the body against external threats; red blood cells, which transport oxygen to tissues and collect waste products from organs and tissues; and platelets, which aid in blood clotting and bleeding prevention. White blood cells and their precursors cannot increase and develop properly in the blood and bone marrow, leading to leukemia, a progressive and malignant illness of the body's hematopoietic organs [1]. These abnormal cells cause symptoms due to bone marrow defects and organ infiltration. They are known as hematopoietic and lymphoid tissue tumors. Due to the complexity of regulation and the differences between hematopoietic cells, it is natural

for these malignancies to be very different. These differences include the predominant cells present in spite, proliferation and apoptosis, clinical features, and their response to treatment. Diagnosis and classification of hematopoietic tissue tumors vary depending on our knowledge of diseases. The accumulation of these cancer cells outside the bone marrow causes masses to form in vital organs of the body, such as the brain, or enlarged lymph nodes, spleen, liver, and dysfunction of vital organs. Identification of leukemia is critical because the prognostic and immediate therapeutic implications are highly dependent on it. Although it is known that automated blood cell image analyzers underestimate the number of blast cells [2], this complex problem has received little attention in the literature.

Manual techniques are primarily used to diagnose cancers. Using these traditional methods has the disadvantage of being time consuming and tedious and requiring an operator with advanced skills to ensure accuracy. As a result, an automated system that is both cost-effective and dependable is always needed. The main technological tools used to detect leukemia cells in the previous decade have been image processing, machine learning, and deep learning techniques [2,3]. Deep learning has turned attention to new categorization models relying on Convolutional Neural Networks (CNNs) [3,4]. In the coming years, automatic categorization algorithms will become a more common aspect of clinical practice in hematological malignancy [5–12]. Various methods based on machine learning models have been investigated for automatically analyzing and detecting leukemia, as will be discussed further below.

Madhloom et al. [13] presented an image-processing-based technique for detecting several forms of leukemia. The accuracy of this procedure was found to be between 85% and 98%. Putzu et al. [14] proposed an image-processing-based technique to distinguish unhealthy from healthy cells in blood and bone marrow samples. Using this method, it was possible to correctly identify 245 of 267 leukocytes (approximately 92%). Nazli-bilek et al. [15] developed a novel approach to automatically calculate and classify white blood cells into five distinct types: basophils, lymphocytes, neutrophils, monocytes, and eosinophils. According to the rotating training set and without PCA, the classifier (NN) had a success rate of 65%. Because the PCA identifies the most significant features of the data vectors in decreasing order, the success rate increased to 95% after applying it to the training set. Habibzadeh et al. [16] presented a methodology for counting and categorizing white blood cells (blasts) in microscopic images into five major classes based on shape, intensity, and texture. The performance of the mentioned system was analyzed using three separate feature sets. It was discovered that DT-CWT, based on multiple image resolution features, achieves the best performance. Boldú et al. [2] developed an acute leukemia diagnosis prediction system based on deep learning. The best architecture for acute leukemia classification was determined by testing VGG16, SENet154, DenseNet121, and ResNet101. Myeloid leukemia had specificity, precision, and sensitivity values of 92.3%, 93.7%, and 100%, respectively. Khandekar et al. [17] introduced an object detection method that uses tiny blood smear pictures to forecast leukemia cells. The MAP (Mean Average Precision) of the ALL-IDB1 dataset was 96.06 percent, whereas the MAP of the C NMC 2019 dataset was 98.7%. Abhishek et al. [18] provided a 500-image dataset of normal, Acute Myeloid Leukemia (AML), and ALL peripheral blood smears. Advanced categorization approaches based on machine learning and deep learning were applied in this research. The aforesaid approaches attained 97% accuracy when the Fully Connected (FC) layers and the final three convolutional layers of VGG16 were fine-tuned for binary classification, and DenseNet121 and SVM obtained an accuracy of 98%. Bibi et al. [19] developed a model that relies on the Internet of Medical Things (IoMT) to enhance and deliver fast and safe detection of leukemia. Based on DenseNet-121 and Residual ResNet-34, the proposed framework identified leukemia subtypes. The findings revealed that the aforementioned models outperformed classical machine learning algorithms in identifying healthy from leukemic subtypes. Rastoqi et al. [20] presented a new two-step method for robust classification of leukocytes for leukemia diagnosis based on a pre-learned network and VGG 16 called LeuFeatx. The accuracy of these researchers' diagnoses based on the

ALL_IDB2 database was 96.15%. Dese et al. [21] presented an automatic diagnosis system based on machine learning to diagnose types of leukemia. Their system was able to classify four common types of leukemia with 97% accuracy. Among the advantages of this research was the access to accuracy above 95%, and the limited scenarios in the experiment can be considered a disadvantage of the study. Chola et al. [22] have used a deep learning framework based on artificial intelligence for fast and automatic identification of blood cells in the classification scenario of eight classes—Basophil, Eosinophil, Erythroblast, Immature Granulocytes, Lymphocyte, Monocyte, Neutrophil, and Platelet. These researchers have compared their model with pre-learned networks such as DenseNet, ResNet, Inception, and MobileNet, and achieved 98% accuracy. Among the advantages of this research was the presentation of the 8-class scenario, and the high computational volume can be considered a disadvantage of this research.

According to previous research, as seen, many papers in recent years have been used to diagnose acute leukemia cells. However, there are challenges in these studies. The first challenge related to previous research, some researchers have used traditional feature selection/extraction algorithms, which require prior knowledge about the subject/problem. Furthermore, the vast majority of these studies have focused on leukemia diagnosis rather than the various types of leukemia. In addition, in most of these studies, no valid database was gathered.

Further to that, the vast majority of these studies lack high diagnostic accuracy. Other research has been developed in recent years based on artificial intelligence and deep learning networks. However, these networks require a lot of data for training. In addition, the deep networks used in previous research include a complex architecture and are often designed in a multi-stage manner and have high computational efficiency, and require expensive hardware. Accordingly, they cannot be used in real-time applications. To overcome the challenges raised, this paper proposes a novel technique for diagnosing various types of acute leukemia, including ALL and AML. A valid database is obtained from the Shahid Ghazi Tabatabai Oncology Center in Tabriz to achieve this goal. A Deep Neural Network (DNN) based on combined GAN and CNN is developed to learn the optimal features. The reason for using the combination of GAN and CNN networks in this work is that the data limitation in training has been solved by using GAN and the CNN classifies acute leukemia cells by using a simple, customized, end-to-end architecture. The findings show an improvement in diagnosis reliability and accuracy, and inference time. The proposed method is capable of making predictions about images that have not been pre-processed in any way. When a person is suspected of having acute leukemia, this strategy provides a decision support system to assist pathologists.

The rest of this work is arranged in the following manner. The background of the CNN and GAN models is first described in Section 2. The developed work framework is detailed in Section 3. Moreover, Section 4 presents the experimental results and compares the proposed method to other approaches. The research conclusions are presented in Section 5 at the end.

## 2. Materials and Methods

A description of the materials and methods used in this study is provided in this section. Accordingly, Section 2.1 discusses the general model of CNN. Section 2.2 explains the available model of GAN.

### 2.1. The General Model of CNN

A standard Artificial Neural Network (ANN) model contains input and output layers and several hidden layers. With CNN, objects can be classified based on context, a characteristic gaining popularity. The three components of a CNN model are the convolution (Conv), the pooling, and the FC [23].

The feature map is generated in the Conv layer by sliding the weight vector, also referred to as the kernel or filter, over the input vector [23]. Convolution operation refers to

the process of sliding the filter both horizontally and vertically. This operation extract $N$ s features from the input image into a single layer representing different features, resulting in $N$ feature maps and $N$ filters.

The exact position of a feature becomes less important once it has been discovered. Therefore, the pooling layer follows the convolution layer. The primary advantage of the pooling strategy is drastically reducing the number of trainable parameters. A few pooling approaches are available, such as average and max pooling, with max pooling being the most widely utilized and reducing feature maps incredibly [23].

The FC layer is the same as the FC network in traditional models. A dot product is computed between the weight vector and the input vector in the FC layer based on the output of the first phase [23].

### 2.2. The General Model of GAN

GANs have become increasingly popular in deep learning over the past few years. GANs can learn the dataset used in experiments and generate a new and real dataset that is not available in the previous dataset. GANs have two significant networks: generators (G) and discriminators (D). A complete opposition exists between these two components. Input data are used to generate noisy images by the G. Creating realistic and natural images is the responsibility of the G. The D is in charge of distinguishing between natural and artificial images. A network D is trained to identify the original data from generated data as accurately as possible. In contrast, the G network has been trained to mislead the D network [24–26].

### 3. Proposed Framework

The overall framework of the developed technique used to diagnose acute leukemia cells consists of three stages: (1) data collection; (2) pre-processing; and (3) a customized CNN model design. Figure 1 depicts the graphical representation of the proposed method framework. In the subsections that follow, each expressed stage is shown in detail.

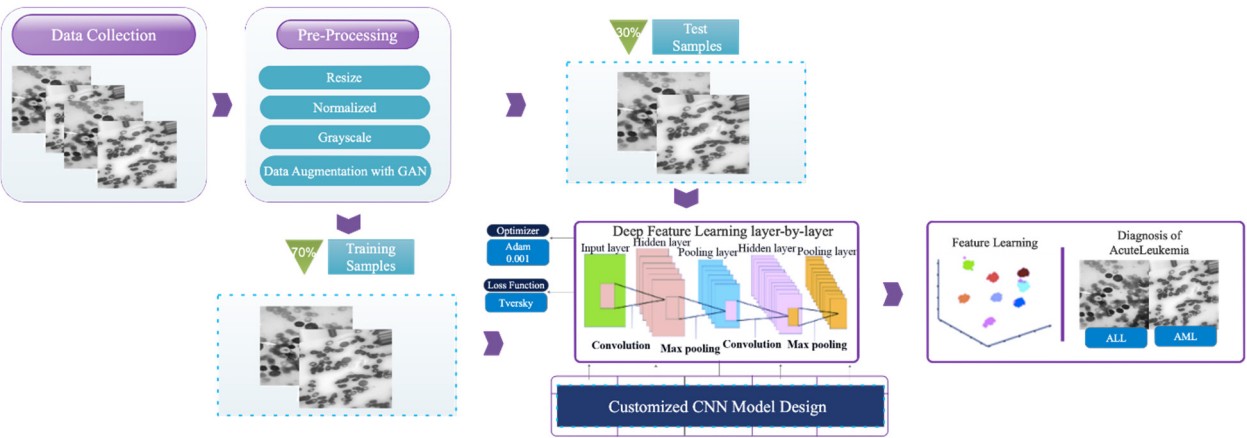

**Figure 1.** The graphical representation of the proposed method framework.

### 3.1. Data Collection

In this study, three steps are taken to collect a dataset. In the first step, clinical symptoms and blood tests are used to separate data on healthy and suspected leukemia patients. Suppose a person's data are suspected of having leukemia. In that case, the number of healthy cells and blast cells in the peripheral blood smear and bone marrow smear is counted in the second step. If there is a minimum chance of leukemia, the person's data are referred to the third step to determine the type of leukemia. In the third step, a doctor labels the properties of lymphocyte and monocyte cells to determine the type of acute leukemia (ALL or AML). The actions related to data collection are shown in Figure 2.

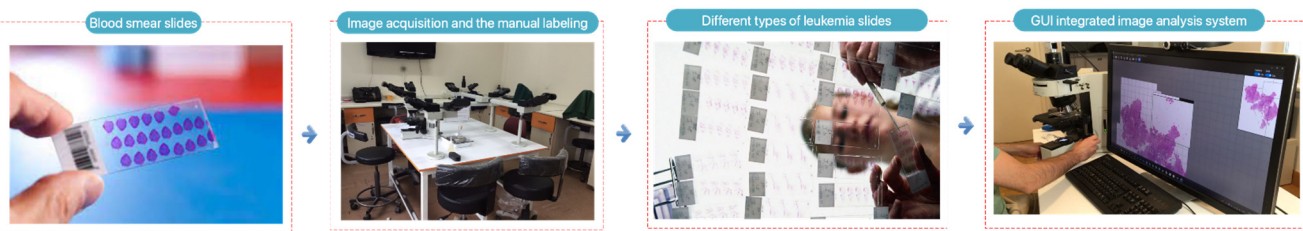

**Figure 2.** How to collect data for leukemia groups.

The images in this study were obtained from the Shahid Ghazi Tabatabai Oncology Center in Tabriz using the three steps described above. This dataset was compiled from 44 patients at various time intervals. 12 of the 44 patients are men, and 32 are women, with an average age of 12 to 70. The dataset from 44 patients included 184 images of ALL and 469 images of AML. On average, there were 5 to 7 usable images from each person. Figure 3 illustrates two different types of acute leukemia.

ALL                                                                                    AML

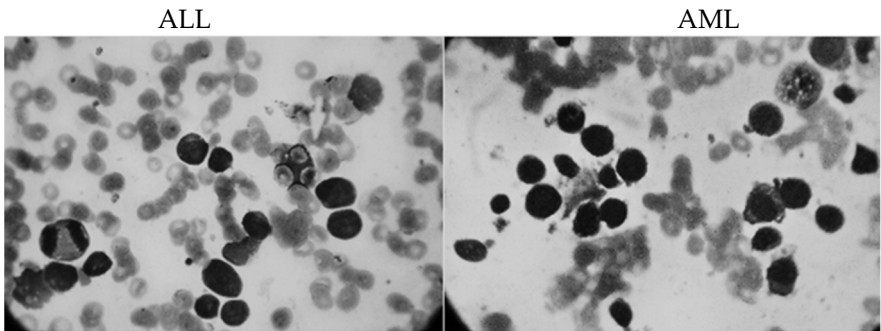

**Figure 3.** Two different types of acute leukemia.

### 3.2. Pre-Processing

The steps required for preprocessing the collected dataset are described in this section. Because the color and format of the collected images vary, they are all resized to $224 \times 224$ and then converted to grayscale to reduce computational volume. They are then normalized to a range of 0 to 1. The number of images in the collected database is unbalanced because the number of images of ALL is limited. As a result, poor classification performance and overfitting issues may arise; to address these issues, the Data Augmentation method (DA), according to the GAN network, is employed to increase ALL images. This process also improves the model's generalization ability. The G network receives as input 100 random entities distributed uniformly and has an output signal of size 50,176 ($224 \times 224$). Four FC layers comprise the G network architecture (256, 512, 1024, and 50,176), with a Batch-Normalization (BN) layer following each. Activation functions Leaky Relu and Tanh are used in this network's hidden layers and at its end. Using an input of size 50,176, the D network makes a decision (whether the images are fake or actual). There are four FC layers (1024, 512, 256, and 1) in this network, followed by a dropout layer at the end of each layer. At the network's end, the sigmoid activation function is used, while the Leaky Relu activation function is used in the hidden layers. The training process in the GAN network is carried out by the Mean Squared Error (MSE) [27] index and the binary cross-entropy optimizer [28], having a 0.0001 learning rate and a batch size of 100 in 1000 epochs. Figure 4 depicts the artificial images created by the GAN for the ALL group. The total number of samples taken before and after DA is 184 and 469, respectively, corresponding to the AML group.

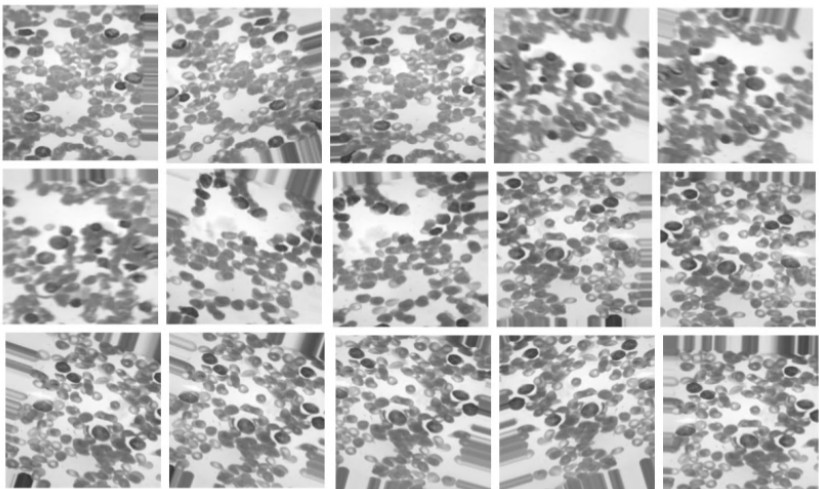

**Figure 4.** The artificial images created by the GAN for the ALL group.

### 3.3. A Customized CNN Model Design

The proposed CNN model includes five maximum-pooling layers and six convolutional layers. The same parameters, including a kernel size of 5 × 5, 20 filters, and max pooling 2 × 2, are used in these layers. Following these layers, four FC layers having 1024, 512, 128, and 2 neurons are added. Seven dropout layers are added after the second, third, fourth, and fifth max-pooling layers, the sixth convolutional layer, and the first and second FC layers, both as a regularization method and to prevent overfitting. Convolutional and FC layers are followed by a Leaky ReLU activation function. Finally, the Softmax activation function is used in the last FC layer to calculate the probabilities of all two classes. Table 1 provides the number of filters, stride size, and architectural details of the customized CNN model. The dimensionality of the hidden layers used for the customized CNN is reduced from 224 × 224 (input size) to 2 (number of classes). The customized CNN model has 820,286 trainable parameters. Figure 5 depicts the architectural details of this customized model.

**Table 1.** The number of filters, stride size, and architectural details of the customized CNN model.

| L | Layer Type | Activation Function | Output Shape | Size of Filter and Pooling | Strides | Number of Filters | Adding |
|---|---|---|---|---|---|---|---|
| 0–1 | Convolution2-D | Leaky ReLU | (None, 20, 112,112) | 5 × 5 | 2 | 20 | **yes** |
| 1–2 | Max-Pooling2-D | - | (None, 20, 56,56) | 2 × 2 | 2 | - | **no** |
| 2–3 | Convolution2-D | Leaky ReLU | (None, 20,56,56) | 5 × 5 | 1 | 20 | **yes** |
| 3–4 | Max-Pooling2-D | - | (None, 20, 28, 28) | 2 × 2 | 2 | - | **no** |
| 4–5 | Convolution2-D | Leaky ReLU | (None, 20, 28, 28) | 5 × 5 | 1 | 20 | **yes** |
| 5–6 | Max-Pooling2-D | - | (None, 20, 14, 14) | 2 × 2 | 2 | - | **no** |
| 6–7 | Convolution2-D | Leaky ReLU | (None, 20, 14, 14) | 5 × 5 | 1 | 20 | **yes** |
| 7–8 | Max-Pooling2-D | - | (None, 20, 7, 7) | 2 × 2 | 2 | - | **no** |
| 8–9 | Convolution2-D | Leaky ReLU | (None, 20, 7, 7) | 5 × 5 | 1 | 20 | **yes** |
| 9–10 | Max-Pooling2-D | | (None, 20, 3, 3) | 2 × 2 | 2 | - | **no** |
| 10–11 | Convolution2-D | Leaky ReLU | (None, 20, 3, 3) | 5 × 5 | 1 | 20 | **yes** |
| 11–12 | Flatten | - | (None, 180) | - | - | - | **-** |
| 12–13 | FC | Leaky ReLU | (None, 1024) | - | - | - | **-** |
| 13–14 | FC | Leaky ReLU | (None, 512) | - | - | - | **-** |
| 14–15 | FC | Leaky ReLU | (None, 128) | - | - | - | **-** |
| 15–16 | FC | Softmax | (None, 2) | - | - | - | **-** |

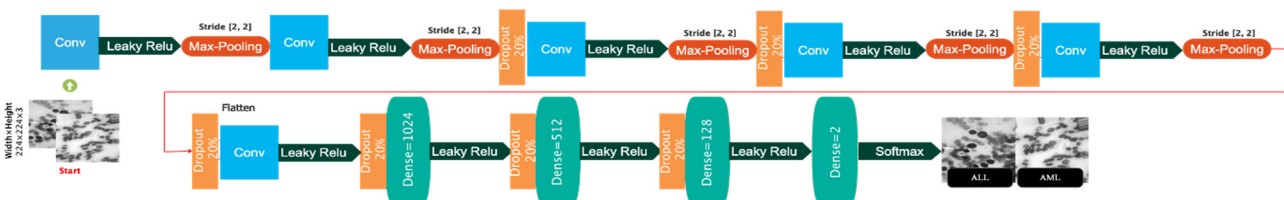

**Figure 5.** The architectural details of the customized CNN model.

All hyper-parameters for the proposed method are carefully adjusted to achieve the best convergence degree. A trial and error method is used to select these parameters. Over the years, a variety of loss functions have been suggested. Typically, loss functions give more weight to foreground voxels to solve the class imbalance problem. The Tversky loss function [29,30] is one such loss function that has been determined experimentally to be more appropriate for delineating biomedical images. The Tversky index [29,30] is defined as follows:

$$S(p, g; \alpha \beta) = \frac{|p \cap g|}{|p \cap g| + \alpha|p/g| + \beta|g/p|} \tag{1}$$

where $g$ and $p$ are the true and predicted labels, respectively. The penalty magnitude of false negatives and false positives is controlled by $\beta$ and $\alpha$. The Tversky loss [29,30] is defined using this index as follows:

$$T(\alpha \beta) = \frac{\sum_{i=1}^{N} p_{0i} g_{0i}}{\sum_{i=1}^{N} p_{0i} g_{0i} + \alpha \sum_{i=1}^{N} p_{0i} g_{1i} + \beta \sum_{i=1}^{N} p_{1i} g_{0i}} \tag{2}$$

In the proposed customized CNN model, the Tversky and Adam with a 0.001 learning rate are used as the loss function and optimizer. The Tversky loss function has a batch size of 100 slices and $\alpha$ and $\beta$ of 0.3 and 0.7, respectively.

Out of the total number of samples collected (938), 657 were randomly used for training (70%), 94 were randomly used for validation (10%), and 187 were randomly used for testing (20%). Figure 6 depicts the allocation of the samples for the training, validation, and test datasets.

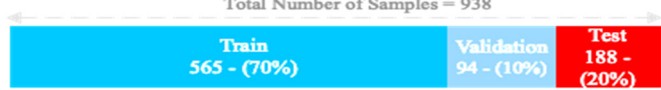

**Figure 6.** The allocation of the samples for the training, validation, and test dataset.

## 4. Experimental Results

An evaluation of the proposed customized CNN model is presented in this section. The models are simulated using the Google Colab [31] cloud service, which has an Nvidia-Tesla T4 GPU and 25 GB of RAM. The accuracy, precision, sensitivity, specificity, kappa, and fscore are presented to investigate the performance of the developed technique [32,33].

Figure 7 depicts the accuracy and error plots of the proposed customized CNN model (i.e., the proposed method) based on training and validation data. It can be seen from Figure 7 that the error of the proposed model decreases with increasing iterations. Furthermore, after 200 iterations, the proposed model achieves an accuracy of 99.5%. Figure 8 shows the ROC plot and confusion matrix based on the proposed customized CNN model for test data. The ROC plot in Figure 8a has a value in the range of 0.9 to 1 and is located in the left hemisphere, indicating that the proposed model is performing as expected. According to Figure 8b, the proposed model's diagnostic performance for different types of acute leukemia, namely AML (class 1) and ALL (class 2), is almost the same.

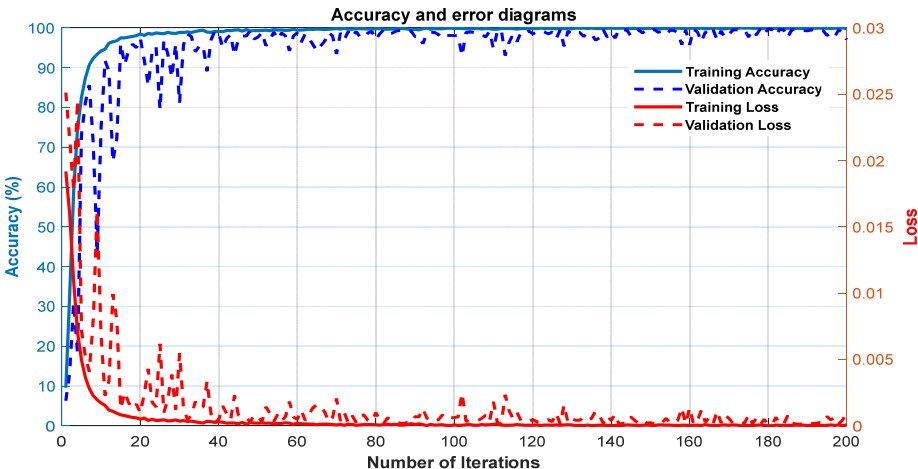

**Figure 7.** The accuracy and error plots of the proposed customized CNN model.

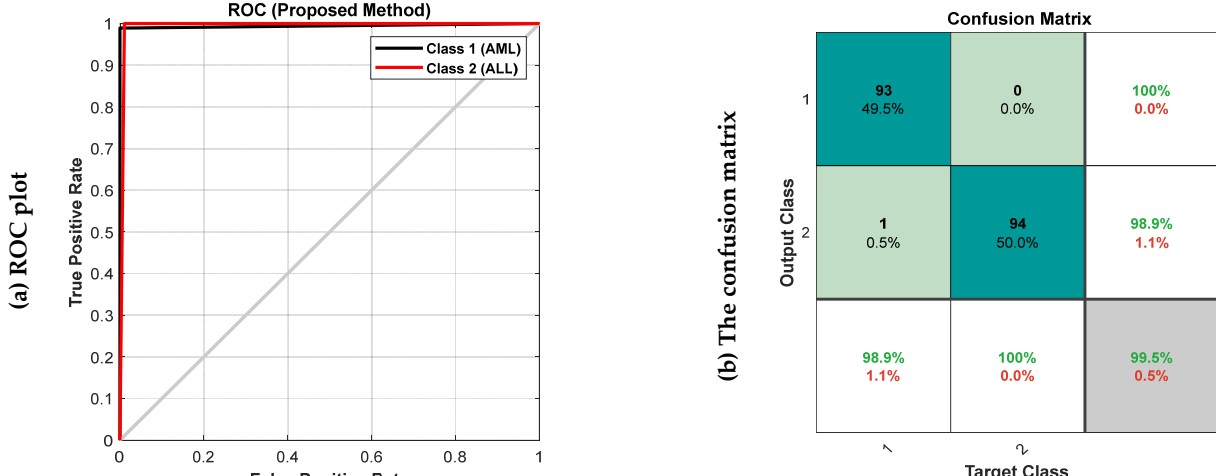

**Figure 8.** The ROC plot and confusion matrix based on the proposed customized CNN model for test data; (**a**) ROC plot, (**b**) The confusion matrix.

Figure 9 shows bar chart plots of accuracy, fscore, kappa, precision, sensitivity, and specificity for test data to further assess the effectiveness of the developed method. The presented customized CNN model has high accuracy, fscore, kappa, precision, sensitivity, and specificity for test data, as shown in Figure 9.

The t-sen, a visual algorithm, is used to further evaluate the proposed method. The purpose of the t-sen is to visualize the separation of samples in different layers of the proposed model. If the samples are separated from each other in the output layer, the efficiency of the proposed network can be proved in the classification of different groups. The t-sen-embedded scatter of test data from the second convolutional layer (Conv 2) and the third FC layer (FC 3) is shown in Figure 10. The third FC layer's t-sen visualization confirms the distinction between two types of acute leukemia, AML and ALL. The developed model can extract desired features from raw images and effectively classify different classes.

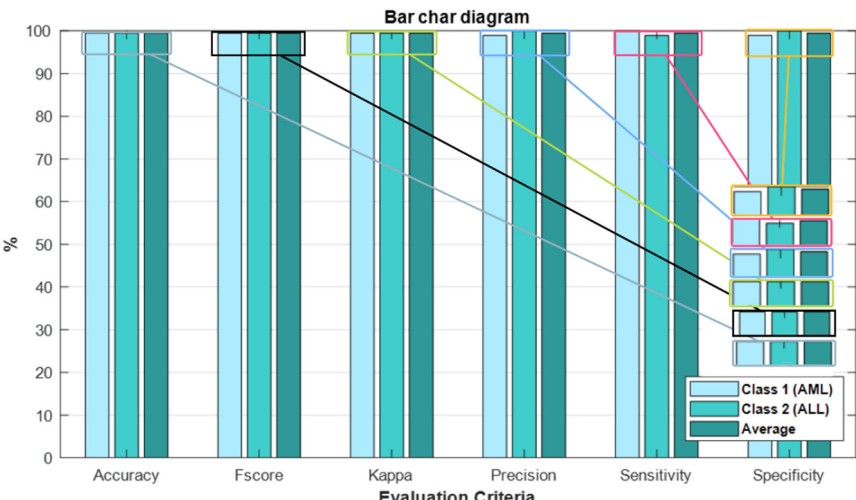

**Figure 9.** The bar chart plots accuracy, fscore, kappa, precision, sensitivity, and specificity based on the proposed customized CNN model.

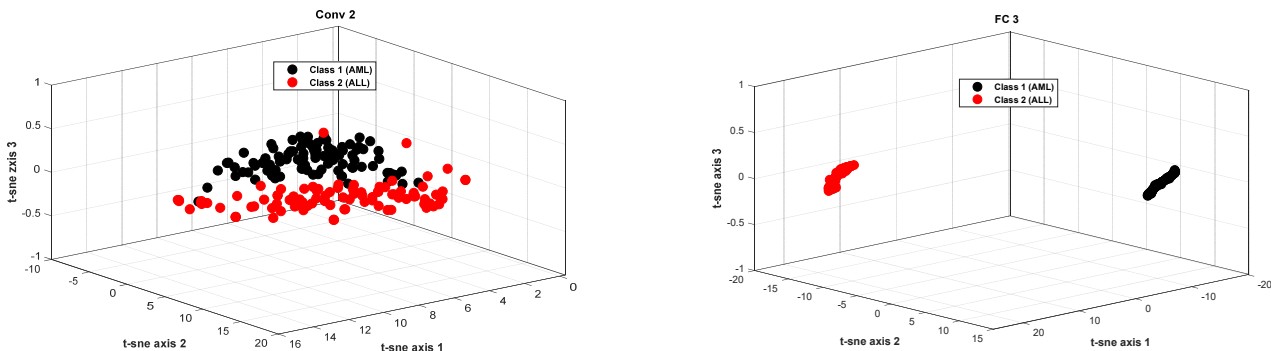

**Figure 10.** The visualization of the test data with the t-sen scatter.

The accuracy and error plots of the proposed customized CNN model with the Tversky loss function (i.e., proposed method) for validation data in different iterations are compared to the CNN model with the cross-entropy and MSE loss functions in Figure 11. Figure 12 compares the accuracy, fscore, kappa, precision, sensitivity, and specificity of the proposed customized CNN model with the Tversky loss function to the CNN model with the cross-entropy and MSE loss functions. It is worth noting that the CNN model employs the architecture shown in Section 3.3. As shown in Figure 11; Figure 12, the proposed customized CNN model with the Tversky loss function (i.e., proposed method) has higher accuracy, fscore, kappa, precision, sensitivity, and specificity, as well as lower loss, than the CNN model with the cross-entropy and MSE loss functions. Therefore, it demonstrates the effectiveness of the proposed approach for diagnosing acute leukemia. Figure 13 shows the confusion matrices and ROC plots for the CNN model with the cross-entropy and MSE loss functions based on test data. As shown in Figure 13, the CNN model with cross-entropy loss function outperforms the CNN model with MSE loss function. It can also be seen from t-sen plots of Figure 10; Figure 13 that the learned features using the proposed customized CNN model with the Tversky loss function are more optimal features than the learned features using the CNN model with the cross-entropy and MSE loss functions, and these features can better differentiate the data of the two classes.

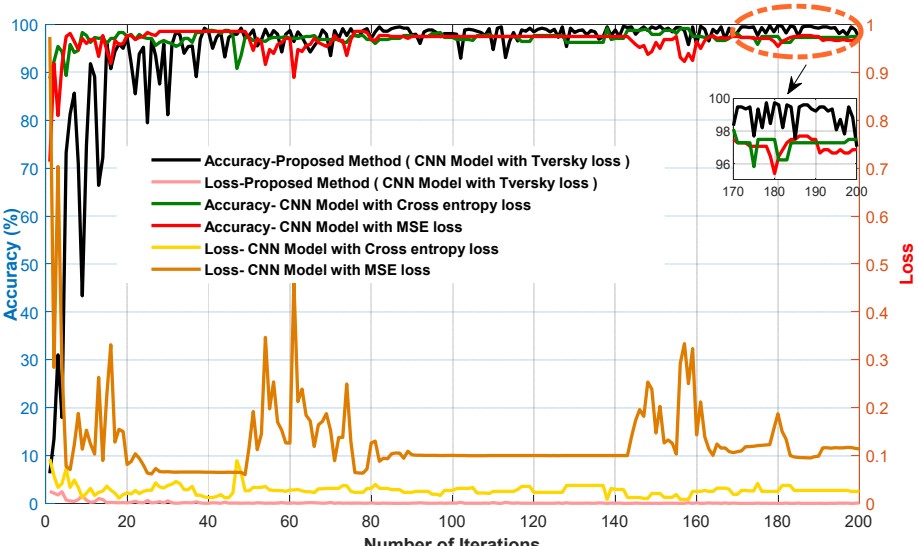

**Figure 11.** The accuracy and error plots of the proposed customized CNN model with the Tversky loss function (i.e., proposed method) compared with the CNN model with the cross-entropy and MSE loss functions.

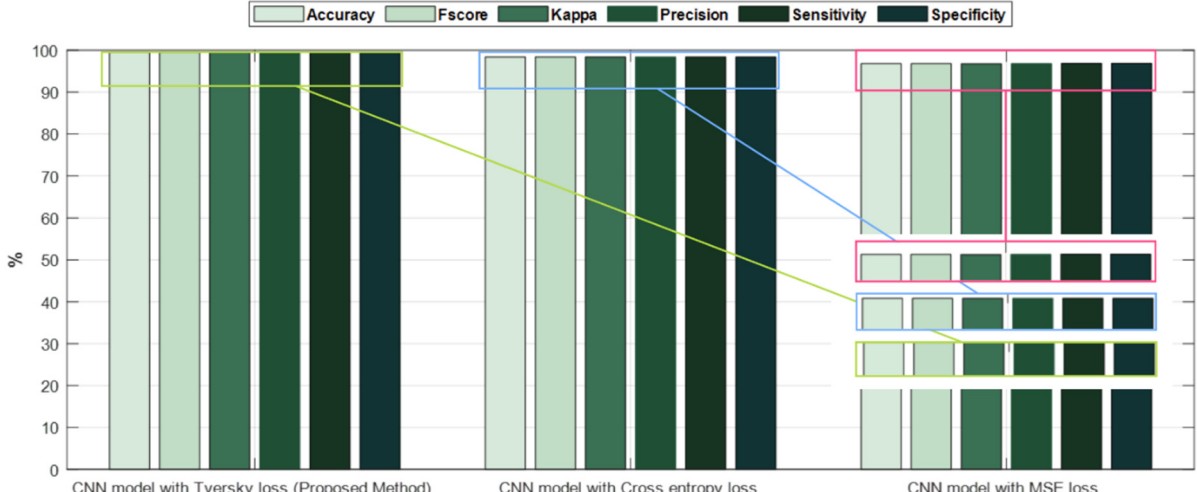

**Figure 12.** Bar chart plots of accuracy, fscore, kappa, precision, sensitivity, and specificity for the proposed customized CNN model with the Tversky loss function compared with the CNN model with the cross-entropy and MSE loss functions.

The 5-fold cross-validation was also performed for the data for more detailed analysis in a customized CNN model. To further research the efficacy of the proposed method, Figure 14 shows the classification accuracy obtained for each fold. As is shown in Figure 14, the accuracy obtained for each fold is approximately higher than 97%, indicating that the overfitting phenomenon did not occur in different folds.

Table 2 presents the leukemia diagnostic accuracy of various methods. As shown in Table 2, most previous studies could only distinguish ALL, whereas the proposed method in this study could diagnose different types of acute leukemia, i.e., AML and ALL. The proposed method has a 99.5% accuracy rate for diagnosing ALL and AML, which is higher than previous studies. Benchmark datasets were used in most previous studies, but a database is gathered in this study. The time-consuming problem of selecting the appropriate method for choosing the features has been solved in this study because the features are extracted hierarchically using an automated manner. However, datasets,

categories, techniques, and simulation environments vary between these studies, making it impossible to compare them one-to-one.

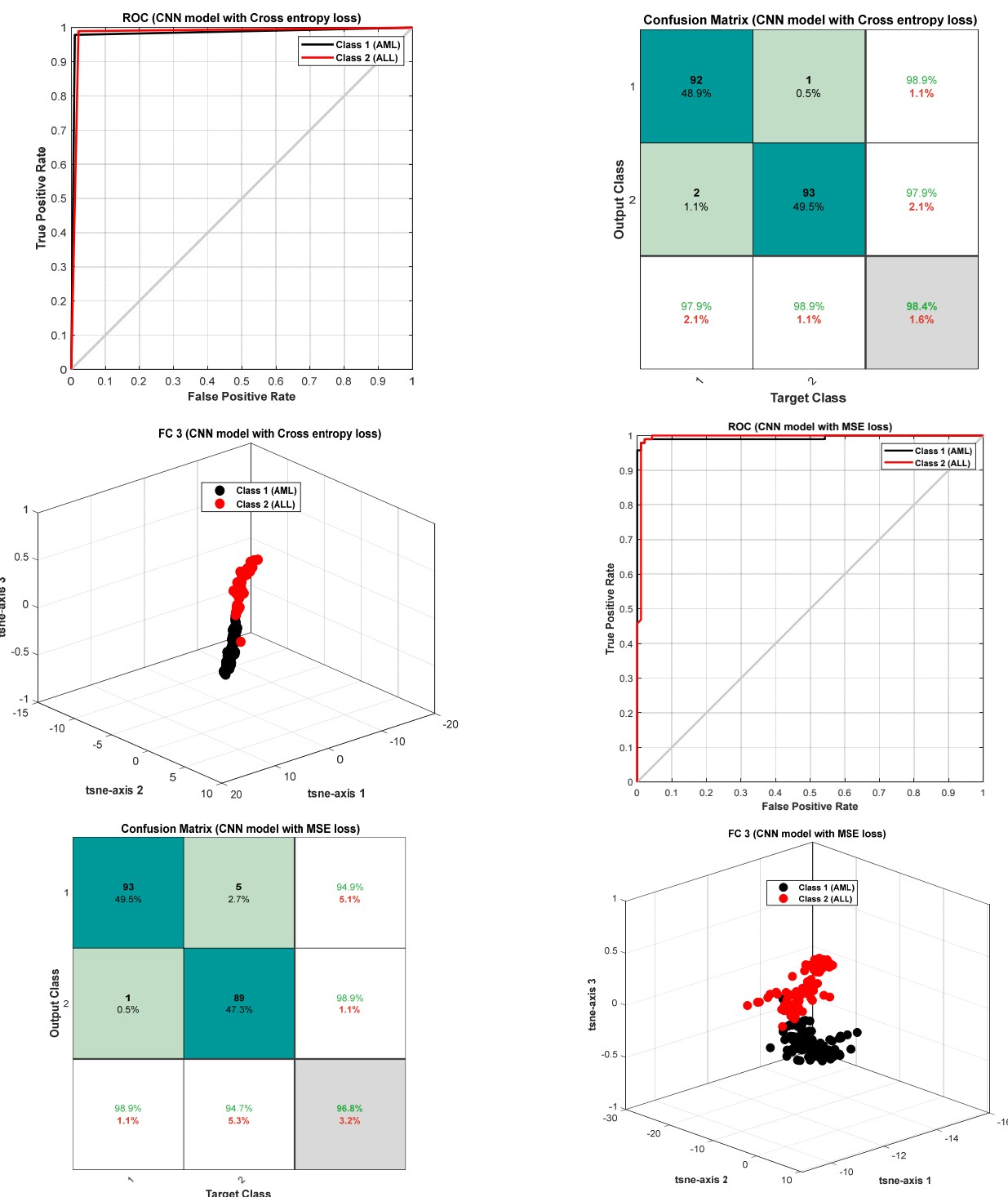

**Figure 13.** The confusion matrices and ROC plots for the CNN model with the cross-entropy and MSE loss functions based on test data.

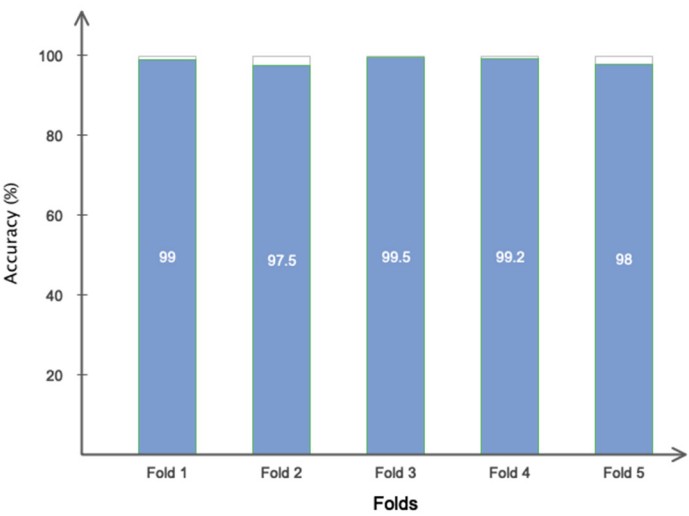

**Figure 14.** The 5-fold cross-validation on data.

**Table 2.** The leukemia diagnostic accuracy of various methods.

| References | Dataset | Classification | Methods | Accuracy |
|---|---|---|---|---|
| Putzu [14] | ALL-IDB1 | ALL | *Image Processing* | 92% |
| Kassanin et al. [34] | ISBI | Healthy and ALL | Customized CNN | 96.17% |
| Agaian et al. [35] | ALL-IDB1 | ALL | Cell Energy Feature with Support Vector Machine | 94% |
| Umamaheswari et al. [36] | ALL-IDB2 | ALL | Customized K-Nearest Neighbor | 96.25% |
| Ahmed et al. [37] | ALL-IDB, ASH Image Bank | Leukemia Subtypes Classification | CNN | 81.74% |
| Al-jaboriy et al. [38] | ALL-IDB1 | ALL | Genetic Algorithm and ANN | 97.07% |
| Nimesh patel et al. [39] | ALL-IDB1 | ALL | SVM | 93.57 |
| Siew chin neoh et al. [40] | ALL-IDB | ALL | SVM and MLP | 96.72 |
| Begum et al. [41] | Not revealed | Leukemia | SVM | Not revealed |
| Fakhouri et al. [42] | Online dataset | Leukemia types | SVM | Not revealed |
| Rdellar et al. [43] | Private dataset | Leukemia types | SVM | 90.3% |
| Chola et al. [22] | HPBC | Leukemia types | BCNet | 98.51%% |
| Rastogi et al. [20] | ALL-IDB2 | ALL-AML | LeuFeatx | 96.15% |
| Proposed Method | Private (ALL-AML) | ALL-AML | Customized CNN | **99.5%** |

## 5. Discussion

To further investigate the proposed performance based on the customized CNN, other popular networks such as Xception [44], VGG19 [45], DeFusionNET [46], and ResNet50 [47] were used for the automatic classification of acute leukemia. The four compared networks have been widely used in acute leukemia studies. Figure 15 shows the accuracy of the proposed model in 150 iterations compared to other networks. According to this figure, as can be seen, the proposed method has been able to obtain the best accuracy among the compared networks for the automatic classification of acute leukemia. However, it converges to the optimal value later. DeFusionNET and ResNet50 networks have similar performance and have been able to achieve accuracy above 90% for classification. However, these networks are oscillating and require many iterations to be stable.

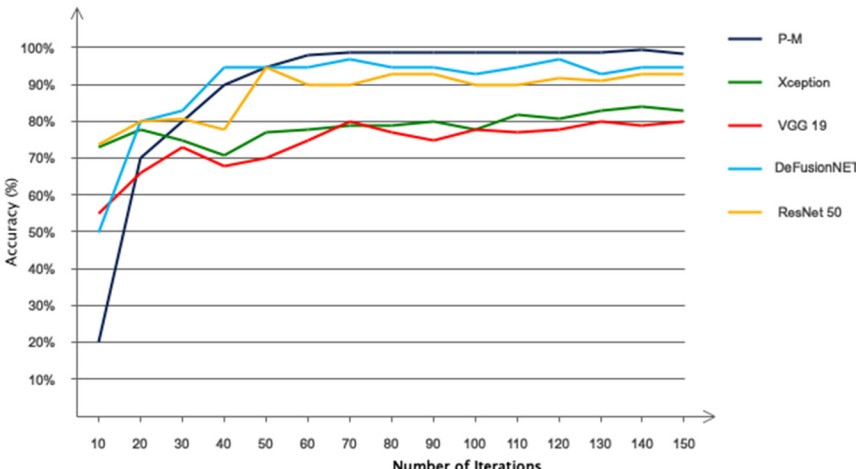

**Figure 15.** Comparing the accuracy of the proposed network with deep transfer learning networks in 150 iterations.

As mentioned in the previous sections, many papers have been presented to diagnose acute leukemia. According to the results of prior research, as seen from Table 2, almost most recent research has reported accuracy above 90% for classification. However, the networks used in the previous study are highly complex and require deep processing units. For example, in the Inception 12 standard network, there are more than ten processing blocks, including more than ten convolutional networks. In comparison, our whole model consists of six convolutional networks. Even without a problematic comparison, the model proposed in this work can also be used in the design of online applications. However, this research, as with other research, has shortcomings. Among the limitations of this research is the limited database, and if you have an extensive database, there will be no need to increase the data by using GAN networks. Classification of two classes and not using multi-class scenarios can be called the following limitation of this research. In the future, we plan to evaluate the relevant research in real-time applications and consider more scenarios for image classification.

## 6. Conclusions

In this paper, the customized CNN model with the Tversky loss function for the diagnosis of acute leukemia cells is presented. The proposed CNN model includes multiple hidden layers, the Adam optimizer, dropout layers, and a learning rate of 0.001, and exhibited an accuracy, fscore, kappa, precision, sensitivity, and specificity above 99%. In addition, the proposed model performs better than the CNN model with the cross-entropy and MSE loss functions. A CNN model based on deep learning significantly improved the diagnosis of acute leukemia cells. Due to the desirable performance of the proposed model, it can be used by doctors and oncology specialists for the automatic diagnosis of acute leukemia in real-time medical applications.

**Author Contributions:** Conceptualization, S.A.; methodology, A.H.N. and A.B.S.; software, S.A. and J.V.G.; validation, S.D. and S.A; writing—original draft preparation, A.H.N. and S.A. All authors have read and agreed to the published version of the manuscript.

**Funding:** This research received no external funding.

**Institutional Review Board Statement:** This study was conducted by the Declaration of Helsinki, and approved by the Ethics Committee of Tabriz University of Medical Sciences. (Protocol code 1399.45.AC and date of approval 1399.4).

**Informed Consent Statement:** Informed consent was obtained from all subjects involved in this study.

**Data Availability Statement:** The data related to this article are publicly available on the GitHub platform under the title Ansari acute leukemia images.

**Conflicts of Interest:** The authors declare no conflict of interest.

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
