# Peer review of "A Customized Efficient Deep Learning Model for the Diagnosis of Acute Leukemia Cells Based on Lymphocyte and Monocyte Images"

_electronics, doi:10.3390/electronics12020322_

Round 1

Reviewer 1 Report

This paper designs a deep-learning model with a customized architecture for detecting acute leukemia using images of lymphocytes and monocytes. In addition the authors presents a novel dataset containing images of Acute Lymphoblastic Leukemia (ALL) and Acute Myeloid Leukemia (AML). The new dataset has been created with the assistance of various experts in order to help the scientific community in its efforts to incorporate machine learning techniques into medical research. In the experiemnts, the proposed model achieved a 99% accuracy rate in diagnosing acute leukemia types, including ALL and AML. The following issues should be addressed before publication.

1. In section 2, the details of CNN and GAN should be compressed since they are commonly known basic concepts;

2. Compared to other image processing and computer vision tasks, what are the challenge of designing deep networks to detect acute leukemia? The authors should give a detailed explanation since this the motivation of the proposed network. In addition, the network structure is very simple, I auggest the authors to cite and discuss some recently proposed network structures, such as: DeFusionNET: Defocus Blur Detection via Recurrently Fusing and Refining Discriminative Multi-scale Deep Features, PAMI 2022

3. I suggest the authors to give an intuitive showing about the detected results in the images.

Author Response

Reviewer#1:

Comments:

This paper designs a deep-learning model with a customized architecture for detecting acute leukemia using images of lymphocytes and monocytes. In addition the authors presents a novel dataset containing images of Acute Lymphoblastic Leukemia (ALL) and Acute Myeloid Leukemia (AML). The new dataset has been created with the assistance of various experts in order to help the scientific community in its efforts to incorporate machine learning techniques into medical research. In the experiemnts, the proposed model achieved a 99% accuracy rate in diagnosing acute leukemia types, including ALL and AML. The following issues should be addressed before publication.

  • ⎫ Thanks to the esteemed reviewer, we believe that your comments have been very useful and effective in enhancing the scientific and writing framework of the manuscript. We have considered all the comments in their entirety and made every effort to correct the manuscript in the manner suggested by the honorable reviewer.

  1. 1. In section 2, the details of CNN and GAN should be compressed since they are commonly known basic concepts;
  • ⎫ The manuscript is revised based on this comment. Based on the request of the respected reviewer, the mathematical background related to CNN and GAN networks has been summarized in section 2.
  1. 2. Compared to other image processing and computer vision tasks, what are the challenge of designing deep networks to detect acute leukemia? The authors should give a detailed explanation since this the motivation of the proposed network. In addition, the network structure is very simple, I auggest the authors to cite and discuss some recently proposed network structures, such as: DeFusionNET: Defocus Blur Detection via Recurrently Fusing and Refining Discriminative Multi-scale Deep Features, PAMI 2022.
  • ⎫ With respect to the reviewer's opinion, the main challenges in previous researches for automatic detection of acute leukemia cells are the use of traditional methods for feature selection/extraction. Using traditional methods for feature selection/extraction requires prior knowledge about the subject/problem. So, the feature selected in one subject or issue may not be optimal in another subject. To put it simply, using these methods will not guarantee the optimality of the feature vector. Also, as it is evident from previous researches, these methods will lead to low diagnosis accuracy. Therefore, implementing an algorithm that learns the correct features corresponding to each case is essential. This will remain as the main benefit of this research. On this basis, this paper proposes a novel technique for diagnosing various types of acute leukemia, including ALL and AML. In order to achieve this goal, a valid database is obtained from the Shahid Ghazi Tabatabai Oncology Center in Tabriz. A Deep Neural Network (DNN) with a customized architecture is developed to learn the optimal features. The findings show an improvement in diagnosis reliability and accuracy and inference time. The proposed method is capable of making predictions about images that have not been pre-processed in any way. When a person is suspected of having acute leukemia, this strategy provides a decision support system to assist pathologists. 
  • ⎫ The opinion of the respected reviewer is absolutely correct, the proposed network architecture seems simple. However, the use of large-size filters in the initial layers and the use of small-size filters in the middle layers has made the network able to extract high-level features from the images and can be more than 95% accurate in two Classify different classes. The reason for using this architecture is that the network can have a high computing speed and can be used in online applications of acute leukemia diagnosis as a medical assistant. 
  • ⎫ Based on the respect of the opinion of the respected reviewer, the proposed network was compared with other deep pre-learned networks:

“To further investigate the proposed performance based on the customized CNN, other popular networks such as Xception, VGG19, DeFusionNET, and ResNet50 were used for the automatic classification of acute leukemia. The four compared networks have been widely used in acute leukemia studies. Figure 14 shows the accuracy of the proposed model in 150 iterations compared to other networks. According to this figure, as can be seen, the proposed method has been able to obtain the best accuracy among the compared networks for the automatic classification of acute leukemia. However, it converges to the optimal value later. DeFusionNET and ResNet50 networks have similar performance and have been able to achieve accuracy above 90% for the purpose of classification. However, these networks are oscillating and require many iterations in order to be stable.

Figure 14. Comparing the accuracy of the proposed network with deep transfer learning networks in 150 iterations.

As mentioned in the previous sections, many papers have been presented in order to diagnose acute leukemia. According to the results of previous research, as can be seen from Table 2, almost most recent research has reported accuracy above 90% for the purpose of classification. However, the networks used in the previous research are extremely complex and require deep processing units. For example, in the Inception 12 standard network, there are more than 10 processing blocks, each of which includes more than 10 convolutional networks. In comparison, our whole model consists of 6 convolutional networks. Even without a complex comparison, it can be concluded that the model proposed in this work can also be used in the design of online applications.”

Which is highlighted in Discussion section, page 13 and lines 347-370.

  1. 3. I suggest the authors to give an intuitive showing about the detected results in the images.
  • ⎫ According to the opinion of the respected reviewer, the visual results for the input samples of the proposed network and their output are presented in Figure 10. According to Figure 12, as can be seen, the proposed network has been able to be effective in separating samples between two classes. According to the figure, as can be seen, almost all samples are separated from each other in the final layer, which is consistent with the results of various evaluations such as accuracy, precision and sensitivity.
  • ⎫ “The t-sne, which is a visual algorithm, is used to evaluate the proposed method further. The purpose of the t-sen is to be able to visualize the separation of samples in different layers of the proposed model. If the samples are separated from each other in the output layer, the efficiency of the proposed network can be proved in the classification of different groups. The t-sne embedded scatter of test data from the second convolutional layer (Conv 2), and the third FC layer (FC 3) is shown in Figure 10. The third FC layer's t-sne visualization confirms the distinction between two types of acute leukemia, AML and ALL. It is evident that the developed model can extract desired features from raw images and effectively classify different classes.”

Figure 10. The visualize of the test data with the t-sne scatter.

Which is highlighted in Result section, pages (8 and 9), and lines 283-291.

Reviewer 2 Report

The purpose of the manuscript is to propose a customized CNN model with a Tversky loss function to optimize parameters and diagnose acute leukemia cells. Authors have used a novel dataset, and preprocessing technology before using CNN to detect. Authors have also claimed to achieve 99.5% accuracy. In terms of the literature survey, presenting the experiment and results – authors need to do more work to make the manuscript suitable for publication.  In most cases, during the experiment, the choices made by the authors seemed experimental and no logical rationale was provided behind the choice. Without the rationale behind the choice of values used here, replicating the technique as a general rule with another dataset would be difficult and model tuning would be very difficult.  The major/minor issues as listed below:

Major/minor points:

1.      Detection and classification of leukemic cells are not novel research, the abstract should have mentioned that extensive research has already been done on this topic and the authors should have also mentioned why they have chosen CNN In one sentence.

2.      The abstract speaks more about the dataset than the technique the authors proposed. Please focus on technique.

3.      The main part of the literature survey line 68 to 140 contains more details than it needs to. Keep the discussion about what the other studies proposed, their success, and shortcomings (if any). It seems the authors went into detail which could have been nice for a review paper. Please keep it short, if readers are interested, they will look up using references.

4.      Since a significant amount of research has been done in the last two years (including 2022), authors are requested to do a better literature survey and include the most recent works as possible. It seems the authors have missed much prominent research on this topic where techniques and results are much better.

5.      Line 142, recent publications do not follow the traditional methods, please do a better literature survey, and use this kind of statement using reference.

6.      Why did the authors not mention numerous uses of CNN already on the same topic in existing research (between lines 142 to 148)?

7.      Line, 147,148 – authors need to explain in a few sentences why they have chosen CNN, and why their technique is novel. Just stating ‘novel’ is not sufficient.

8.      Please keep section 2 short since readers can look it up using references. These are all published materials from other sources, and they are taking up space without adding any new information.  

9.      Please include a flow diagram showing the steps of data collection. Please provide database information in detail (age range, preexisting conditions, how many images per person, etc.)

10.   How did the authors choose the parameter for GAN (mentioned between lines 251 – 262)?

11.   How did the authors choose the parameter for CNN (mentioned between lines 269 – 280)?

12.   What is the rationale for choosing the Tversky loss function to optimize parameters in CNN? Any selection of hyperparameter or loss function cannot be random and needs to be rationalized.

13.   Equations 6-10 are not necessary; they are very well-known performance metrics. The use of reference is sufficient.

14.   Did authors use cross-validation while calculating performance metrics?

15.   Figure 9 and 11, start from 60 or 70 on the Y axis so that readers can get the difference at the top.

16.   What is the purpose of figure 10? Explain in text.

17.   Authors are requested to revise table 2 using information and updated research from the following review paper or similar other paper: https://www.sciencedirect.com/science/article/pii/S1350453321001168?casa_token=cvKFzj_yXvMAAAAA:moZHFEnx8Sd2HSTWcSUWvKUDle6HFd23WXUBEiBuKDCKvfhWWAVSh_2dl1powQiB3Wfkmpvmlg

18.   Authors need to include a section called discussion to discuss the performance of their technique compared with other techniques in terms of why and how. Not just numbers.

19.   Authors need to include a section or subsection dedicated to the limitations or constraints of their experiments and probable ways to solve those issues (the limitations they have provided are at the end part of the conclusion, which is not the place they should have been).

Minor points:

1.      Authors need to make sure they have provided citations and taken permission from all the images from other sources.  

Author Response

: a.habibzad@srbiau.ac.ir

Reviewer#2:

Comments:

The purpose of the manuscript is to propose a customized CNN model with a Tversky loss function to optimize parameters and diagnose acute leukemia cells. Authors have used a novel dataset, and preprocessing technology before using CNN to detect. Authors have also claimed to achieve 99.5% accuracy. In terms of the literature survey, presenting the experiment and results – authors need to do more work to make the manuscript suitable for publication.  In most cases, during the experiment, the choices made by the authors seemed experimental and no logical rationale was provided behind the choice. Without the rationale behind the choice of values used here, replicating the technique as a general rule with another dataset would be difficult and model tuning would be very difficult.  The major/minor issues as listed below:

  • ⎫ Thanks to the esteemed reviewer, we believe that your comments have been very useful and effective in enhancing the scientific and writing framework of the manuscript. We have considered all the comments in their entirety and made every effort to correct the manuscript in the manner suggested by the honorable reviewer.

  1. 1. Detection and classification of leukemic cells are not novel research, the abstract should have mentioned that extensive research has already been done on this topic and the authors should have also mentioned why they have chosen CNN In one sentence.
  • ⎫ The manuscript is revised based on this comment. According to the reviewer's opinion, the reason for using CNN to diagnose acute leukemia has been added in the abstract “Abstract: The production of blood cells is affected by leukemia, a type of bone marrow cancer or blood cancer. Deoxyribonucleic acid (DNA) related to immature cells, particularly white cells, and is damaged in a variety of ways in this disease. When a radiologist is involved in diagnosing acute leukemia cells, the diagnosis is time-consuming and does not provide good accuracy. For this purpose, many researches have been conducted for automatic diagnosis of acute leukemia. However, these studies have low detection speed and accuracy. Machine learning and artificial intelligence techniques are now playing an important role in medical sciences, particularly in the detection and classification of leukemic cells. These methods assist doctors in detecting dis-eases earlier, reducing their workload and the possibility of errors. This research aims to design a deep-learning model with a customized architecture for detecting acute leukemia using images of lymphocytes and monocytes. This study presents a novel dataset containing images of Acute Lymphoblastic Leukemia (ALL) and Acute Myeloid Leukemia (AML). The new dataset has been created with the assistance of various experts in order to help the scientific community in its efforts to incorporate machine learning techniques into medical research. Increasing the scale of the dataset is achieved with a Generative Adversarial Network (GAN). The proposed CNN model based on the Tversky loss function, includes 6 convolution layers, 4 dense layers, and finally Softmax activation function for the classification of acute leukemia images. The proposed model achieved a 99% accuracy rate in diagnosing acute leukemia types, including ALL and AML. Compared to previous researches, the proposed network provides a promising performance in terms of speed and accuracy, and based on the results, the proposed model can be used as an assistant to doctors and specialists in practical applications”.

Which is highlighted in abstract section, page 1 and lines 15-30.

  1. 2. The abstract speaks more about the dataset than the technique the authors proposed. Please focus on technique.
  • ⎫ The manuscript is revised based on this comment. “The production of blood cells is affected by leukemia, a type of bone marrow cancer or blood cancer. Deoxyribonucleic acid (DNA) related to immature cells, particularly white cells, and is damaged in a variety of ways in this disease. When a radiologist is involved in diagnosing acute leukemia cells, the diagnosis is time-consuming and does not provide good accuracy. For this purpose, many researches have been conducted for automatic diagnosis of acute leukemia. However, these studies have low detection speed and accuracy. Machine learning and artificial intelligence techniques are now playing an important role in medical sciences, particularly in the detection and classification of leukemic cells. These methods assist doctors in detecting dis-eases earlier, reducing their workload and the possibility of errors. This research aims to design a deep-learning model with a customized architecture for detecting acute leukemia using images of lymphocytes and monocytes. This study presents a novel dataset containing images of Acute Lymphoblastic Leukemia (ALL) and Acute Myeloid Leukemia (AML). The new dataset has been created with the assistance of various experts in order to help the scientific community in its efforts to incorporate machine learning techniques into medical research. Increasing the scale of the dataset is achieved with a Generative Adversarial Network (GAN). The proposed CNN model based on the Tversky loss function, includes 6 convolution layers, 4 dense layers, and finally Softmax activation function for the classification of acute leukemia images. The proposed model achieved a 99% accuracy rate in diagnosing acute leukemia types, including ALL and AML. Compared to previous researches, the proposed network provides a promising performance in terms of speed and accuracy, and based on the results, the proposed model can be used as an assistant to doctors and specialists in practical applications.”

Which is highlighted in abstract section, page 1 and lines 15-30.

  1. 3. The main part of the literature survey line 68 to 140 contains more details than it needs to. Keep the discussion about what the other studies proposed, their success, and shortcomings (if any). It seems the authors went into detail which could have been nice for a review paper. Please keep it short, if readers are interested, they will look up using references.
  • ⎫ The manuscript is revised based on this comment. Yes, the opinion of the respected reviewer is completely correct. According to the opinion of the respected referee, the review section on the previous researches was summarized and only the main concepts, advantages, and disadvantages of each research remained.

“Manual techniques are primarily used to diagnose cancers. Using these traditional methods has the disadvantage of being time-consuming and tedious, as well as requiring an operator with advanced skills to ensure accuracy. As a result, an automated system that is both cost-effective and dependable is always required.  The main technological tools used to detect leukemia cells in the previous decade have been image processing, machine learning, and deep learning techniques [2-3]. Deep learning has turned attention to new categorization models relying on Convolutional Neural Networks (CNNs) [3-4]. In the coming years, automatic categorization algorithms will become a more common aspect of clinical practice in the field of hematological malignancy [5-12]. Various methods based on machine learning models have been investigated to date for automatically analyzing and detecting leukemia, as will be discussed further below.

Madhloom et al. [13] presented an image-processing-based technique for detecting several forms of leukemia. The accuracy of this procedure was found to be between 85% and 98%. Putzu et al. [14] proposed an image-processing-based technique to distinguish unhealthy from healthy cells in blood and bone marrow samples.  It was possible to correctly identify 245 of 267 leukocytes by using this method (about 92%). Nazlibilek et al. [15] developed a novel approach to automatically calculate and classify white blood cells into five distinct types: basophils, lymphocytes, neutrophils, monocytes, and eosinophils.  According to the rotating training set and without the use of PCA, the classifier (NN) had a success rate of 65%. Because the PCA identifies the most significant features of the data vectors in decreasing order, the success rate increased to 95% after applying it to the training set. Habibzadeh et al. [16] presented a methodology for counting and categorizing white blood cells (blasts) in microscopic images into five major classes based on shape, intensity, and texture. The performance of the mentioned system was analyzed using three separate feature sets, and it was discovered that DT-CWT, which is based on multiple image resolution features, achieves the best performance. Boldú et al. [2] developed an acute leukemia diagnosis prediction system based on deep learning.  The best architecture for acute leukemia classification was determined by testing VGG16, SENet154, DenseNet121, and ResNet101. Myeloid leukemia had specificity, precision, and sensitivity values of 92.3%, 93.7%, and 100%, respectively. Khandekar et al. [17] introduced an object detection method that uses tiny blood smear pictures to forecast leukemia cells. The MAP (Mean Average Precision) of the ALL-IDB1 dataset was 96.06 percent, whereas the MAP of the C NMC 2019 dataset was 98.7%. Abhishek et al. [18] provided a 500-image dataset of normal, Acute Myeloid Leukemia (AML), and ALL peripheral blood smears. Advanced categorization approaches based on machine learning and deep learning were applied in this research. The aforesaid approaches attained 97% accuracy when the Fully Connected (FC) layers and the final three convolutional layers of VGG16 were fine-tuned for binary classification, and DenseNet121 and SVM obtained an accuracy of 98%. Bibi et al. [19] developed a model that relies on the Internet of Medical Things (IoMT) to enhance and de-liver fast and safe detection of leukemia. Based on DenseNet-121 and Residual ResNet-34, the proposed framework identified subtypes of leukemia. The findings revealed that the aforementioned models outperformed classical machine learning algorithms in identifying healthy from leukemic subtypes. Rastoqi et al. [20] presented a new two-step method for robust classification of leukocytes for leukemia diagnosis based on pre-learned network and VGG 16 called LeuFeatx. The accuracy of these researchers' diagnosis based on the ALL_IDB2 database was 96.15%. Dese et al. [21] presented an automatic diagnosis system based on machine learning to diagnose types of leukemia. Their system was able to classify four common types of leukemia with 97% accuracy. One of the advantages of this research was the access to accuracy above 95%, and the limited scenarios in the experiment can be considered as a disadvantage of the research. Chola et al. [22] have used a deep learning framework based on artificial intelligence for fast and automatic identification of blood cells in the classification scenario of 8 classes Basophil, Eosinophil, Erythroblast, Immature Granulocytes, Lymphocyte, Monocyte, Neutrophil, and Platelet. These researchers have compared their model with pre-learned networks such as DenseNet, ResNet, Inception, MobileNet, and achieved 98% accuracy. One of the advantages of this research was the presentation of the 8-class scenario, and the high computational volume can be considered as a disadvantage of this research.”

Which is highlighted in introduction section, page 2 and lines 57-113.

  1. 4. Since a significant amount of research has been done in the last two years (including 2022), authors are requested to do a better literature survey and include the most recent works as possible. It seems the authors have missed much prominent research on this topic where techniques and results are much better.
  • ⎫ The manuscript is revised based on this comment. The opinion of the respected reviewer, 3 new studies published in 2022 were added to the review section of previous studies and the advantages and disadvantages of each were examined. 

“Rastoqi et al. [20] presented a new two-step method for robust classification of leukocytes for leukemia diagnosis based on pre-learned network and VGG 16 called LeuFeatx. The accuracy of these researchers' diagnosis based on the ALL_IDB2 database was 96.15%.

Chola et al. [21] have used a deep learning framework based on artificial intelligence for fast and automatic identification of blood cells in the classification scenario of 8 classes Basophil, Eosinophil, Erythroblast, Immature Granulocytes, Lymphocyte, Monocyte, Neutrophil, and Platelet. These researchers have compared their model with pre-learned networks such as DenseNet, ResNet, Inception, MobileNet, and achieved 98% accuracy. One of the advantages of this research was the presentation of the 8-class scenario, and the high computational volume can be considered as a disadvantage of this research.

Dese et al. [22] presented an automatic diagnosis system based on machine learning to diagnose types of leukemia. Their system was able to classify four common types of leukemia with 97% accuracy. One of the advantages of this research was the access to accuracy above 95%, and the limited scenarios in the experiment can be considered as a disadvantage of the research.”

Which is highlighted in introduction section, page 3 and lines 99-113.

  1. 5. Line 142, recent publications do not follow the traditional methods, please do a better literature survey, and use this kind of statement using reference.
  • ⎫ The manuscript is revised based on this comment. Based on the opinion of the respected reviewer, the relevant text was modified and organized as follows:

“According to previous research, as seen, many papers in recent years have been used to diagnose acute leukemia cells. However, there are challenges in these studies. The first challenge related to previous research, some researchers have used traditional feature selection/extraction algorithms, which require prior knowledge about the subject/problem. Furthermore, the vast majority of these studies have focused on leukemia diagnosis rather than the various types of leukemia. Besides that, in the majority of these studies, no valid database was gathered. Further to that, the vast majority of these studies lack high diagnostic accuracy. Other research has been developed in recent years based on artificial intelligence and deep learning networks. However, these networks require a lot of data for training. In addition, the deep networks used in previous research include a complex architecture and are often designed in a multi-stage manner and have high computational efficiency, and require expensive hardware. Accordingly, they cannot be used in real-time applications. In order to overcome the challenges raised, this paper proposes a novel technique for diagnosing various types of acute leukemia, including ALL and AML. In order to achieve this goal, a valid database is obtained from the Shahid Ghazi Tabatabai Oncology Center in Tabriz. A Deep Neural Network (DNN) with a customized architecture (end-to-end) is developed to learn the optimal features. The findings show an improvement in diagnosis reliability and accuracy and inference time. The proposed method is capable of making predictions about images that have not been pre-processed in any way. When a person is suspected of having acute leukemia, this strategy provides a decision support system to assist pathologists.”

Which is highlighted in introduction section, page 3 and lines 114-136.

  1. 6. Why did the authors not mention numerous uses of CNN already on the same topic in existing research (between lines 142 to 148)?
  • ⎫ The manuscript is revised based on this comment. According to the opinion of the respected reviewer, as mentioned in comment 4, 3 new studies published in 2022 to detect acute leukemia cells based on deep learning networks have been added to the review section of previous studies and the advantages and disadvantages of each were investigated. Also, the existing challenges to detect acute leukemia cells were corrected based on recent articles based on deep learning networks.

Which is highlighted in introduction section, page 3 and lines 114-136.

  1. 7. Line, 147,148 – authors need to explain in a few sentences why they have chosen CNN, and why their technique is novel. Just stating ‘novel’ is not sufficient.
  • ⎫ The manuscript is revised based on this comment. According to the opinion of the respected reviewer, the reason for using CNN is explained in the relevant section as follows: 

“A Deep Neural Network (DNN) based on combined GAN and CNN is developed to learn the optimal features. The reason for using the combination of GAN and CNN networks in this work is that the data limitation in training has been solved by using GAN and the CNN classifies acute leukemia cells by using a simple, customized, end-to-end architecture.”

Which is highlighted in introduction section, page 2 and lines 129-133.

  1. 8. Please keep section 2 short since readers can look it up using references. These are all published materials from other sources, and they are taking up space without adding any new information.  
  • ⎫ The manuscript is revised based on this comment. According to the opinion of the respected reviewer, section 2, which examines the mathematical background of CNN and GAN networks, has been summarized:

“A description of the materials and methods used in this study is provided in this section. Accordingly, Section 2.1 discusses the general model of CNN. Section 2.2 explains the general model of GAN.

2.1. General Model of CNN

A standard Artificial Neural Network (ANN) model contains input and output layers, as well as several hidden layers. With CNN, objects can be classified based on context, a characteristic that is gaining popularity. The three components of a CNN model are the convolution (Conv), the pooling, and the FC [20]. 

The feature map is generated in the Conv layer by sliding the weight vector, also referred to as the kernel or filter, over the input vector [20]. Convolution operation refers to the process of sliding the filter both horizontally and vertically. This operation extracts features from the input image into a single layer that represents different features, resulting in feature maps and filters. 

The exact position of a feature becomes less important once it has been discovered. Therefore, the pooling layer follows the convolution layer. The primary advantage of the pooling strategy is drastically reducing the number of trainable parameters. There are a few pooling approaches available, such as average and max-pooling, with max-pooling being the most widely utilized and reducing feature maps greatly [20]. 

The FC layer is the same as the FC network in traditional models. A dot product is computed between the weight vector and the input vector in the FC layer based on the output of the first phase [20].

2.2. General Model of GAN

During the past few years, GANs have become increasingly popular in the field of deep learning. GANs can learn the dataset used in experiments and generate a new and real dataset that is not available in the previous dataset. GANs consist of two major net-works: a generator (G) and a discriminator (D). A complete opposition exists between these two components. Input data is used to generate noisy images by the G. Creating realistic and natural images is the responsibility of the G. The D is in charge of distinguishing between real and artificial images. A network D is trained to identify the original data from generated data as accurately as possible. In contrast, the G network has been trained to mislead the D network [22, 23]. Figure 1 depicts a graphical representation of GANs.”

Which is highlighted in section 2.1, page 3 and lines 143-173.

  1. 9. Please include a flow diagram showing the steps of data collection. Please provide database information in detail (age range, preexisting conditions, how many images per person, etc.)
  • ⎫ The manuscript is revised based on this comment. Based on the reviewer's opinion, the details related to the database were added to section 2.1. Also, the block diagram under the title of how to collect data was added to the relevant section.

Figure 2. How to collect data for leukemia groups.

Which is highlighted in section 3.1, page 5.

  1. 10. How did the authors choose the parameter for GAN (mentioned between lines 251 – 262)?
  • ⎫ The manuscript is revised based on this comment. The proposed GAN network is organized based on trial and error. In the proposed GAN architecture, the generative network takes random vectors of 100 from a uniform distribution as input and outputs a signal of size 50176 (224 × 224). The network architecture consists of four fully connected dense layers (256, 512, 1024, and 50,176), each layer followed by a batch normalization layer. Leaky-Relu is used as the activation function in the hidden layers and the tanh activation function is used at the end of the network. The discriminator network takes an input of size 50176 and outputs a decision (if the images are real or fake). In this network, four fully connected dense layers (1024, 512, 256, and 1) are used, and each layer is followed by a dropout layer. In the proposed GAN, Leaky-Relu is used as the activation function in the hidden layers and the sigmoid activation function at the end of the network. The training process in the GAN network is carried out by the Mean Squared Error (MSE) [24] index and the binary cross-entropy optimizer [25], having a 0.0001 learning rate and a batch size of 100 in 1000 epochs.

Which is highlighted in section 3.2, pages 209-223.

  1. 11. How did the authors choose the parameter for CNN (mentioned between lines 269 – 280)?
  • ⎫ The manuscript is revised based on this comment. All hyper-parameters for the proposed method are carefully adjusted to achieve the best convergence degree. A trial and error method is used to select these parameters. Over the years, a variety of loss functions have been suggested. Typically, loss functions give more weight to foreground voxels in order to solve the class imbalance problem. The Tversky loss function [26-27] is one such loss function that has been determined experimentally to be more appropriate for delineating biomedical images. The Tversky index [26-27] is defined as follows:

(1)

Where  and  are the true and predicted labels, respectively. The penalty magnitude of false negatives and false positives is controlled by  and . The Tversky loss [26-27] is defined using this index as follows:

(2)

In the proposed customized CNN model, the Tversky and Adam with a 0.001 learning rate are used as the loss function and optimizer. The Tversky loss function has a batch size of 100 slices and  and  of 0.3 and 0.7, respectively. 

Which is highlighted in section 3.3, page 6.

  1. 12. What is the rationale for choosing the Tversky loss function to optimize parameters in CNN? Any selection of hyper parameter or loss function cannot be random and needs to be rationalized.
  • ⎫ This loss function has been used in other studies [a, b] in the field of image processing. Accordingly, this loss function was also used in the present study and its performance was compared with other cost functions such as MSE and Cross Entropy (fig 11).

[a] Stefano, A.; Comelli, A. Customized efficient neural network for covid-19 infected region identification in ct images. Jour-nal of Imaging 2021, 7, 131.

[b] Comelli, A.; Dahiya, N.; Stefano, A.; Vernuccio, F.; Portoghese, M.; Cutaia, G.; Bruno, A.; Salvaggio, G.; Yezzi, A. Deep learning-based methods for prostate segmentation in magnetic resonance imaging. Applied Sciences 2021, 11, 782.

  1. 13. Equations 6-10 are not necessary; they are very well-known performance metrics. The use of reference is sufficient.
  • ⎫ The manuscript is revised based on this comment. According to the opinion of the respected referee, formulas 6 to 11 have been removed from the manuscript.

  1. 14. Did authors use cross-validation while calculating performance metrics?
  • ⎫ The manuscript is revised based on this comment. According to the reviewer's opinion, a 5-fold cross-validation was performed on the data.

“The 5-fold cross-validation was also performed for all of the data for a more detailed analysis. To further research the efficacy of the proposed method, Figure 14 shows the classification accuracy obtained for each fold. As is shown in Figure 14, the accuracy obtained for each fold is approximately higher than 98%, indicating that the overfitting phenomenon did not occur in different folds.”

Figure 14. The 5-fold cross-validation on data.

Which is highlighted in Result section, page 11 and lines 328-335.

  1. 15. Figure 9 and 11, start from 60 or 70 on the Y axis so that readers can get the difference at the top.
  • ⎫ The manuscript is revised based on this comment. According to the opinion of the respected referee, in order to clarify the results of each diagram, a magnifying glass was used in the diagrams. 

Which are highlighted in figs 9 and 11.

  1. 16. What is the purpose of figure 10? Explain in text.
  • ⎫ The manuscript is revised based on this comment. The t-sne, which is a visual algorithm, is used to evaluate the proposed method further. The purpose of the t-sen is to be able to visualize the separation of samples in different layers of the proposed model. If the samples are separated from each other in the output layer, the efficiency of the proposed network can be proved in the classification of different groups. The t-sne embedded scatter of test data from the second convolutional layer (Conv 2), and the third FC layer (FC 3) is shown in Figure 10. The third FC layer's t-sne visualization confirms the distinction between two types of acute leukemia, AML and ALL. It is evident that the developed model can extract desired features from raw images and effectively classify different classes.

Which is highlighted in results section, page 8 , lines 283-291.

  1. 17. Authors are requested to revise table 2 using information and updated research from the following review paper or similar other paper: https://www.sciencedirect.com/science/article/pii/S1350453321001168?casa_token=cvKFzj_yXvMAAAAA:moZHFEnx8Sd2HSTWcSUWvKUDle6HFd23WXUBEiBuKDCKvfhWWAVSh_2dl1powQiB3Wfkmpvmlg
  • ⎫ The manuscript is revised based on this comment. According to the opinion of the respected reviewer, recent researches have been added to Table 2 for the purpose of comparison according to the introduced article. 

Which is highlighted table 2.

  1. 18. Authors need to include a section called discussion to discuss the performance of their technique compared with other techniques in terms of why and how. Not just numbers.
  • ⎫ The manuscript is revised based on this comment. Yes, the opinion of the respected reviewer is completely correct. According to the opinion of my respected reviewer, a section under the title of discussion was considered in the article and the proposed method was compared with other researches in terms of performance. Also, the advantages and disadvantages of the proposed method along with suggestions for future research were transferred to this section.

“To further investigate the proposed performance based on the customized CNN, other popular networks such as Xception [44], VGG19 [45], DeFusionNET [46], and ResNet50 [47] were used for the automatic classification of acute leukemia. The four compared networks have been widely used in acute leukemia studies. Figure 14 shows the accuracy of the proposed model in 150 iterations compared to other networks. According to this figure, as can be seen, the proposed method has been able to obtain the best accuracy among the compared networks for the automatic classification of acute leukemia. However, it converges to the optimal value later. DeFusionNET and ResNet50 networks have similar performance and have been able to achieve accuracy above 90% for the purpose of classification. However, these networks are oscillating and require many iterations in order to be stable.

Figure 14. Comparing the accuracy of the proposed network with deep transfer learning networks in 150 iterations.

As mentioned in the previous sections, many papers have been presented in order to diagnose acute leukemia. According to the results of previous research, as can be seen from Table 2, almost most recent research has reported accuracy above 90% for the purpose of classification. However, the networks used in the previous research are extremely complex and require deep processing units. For example, in the Inception 12 standard network, there are more than 10 processing blocks, each of which includes more than 10 convolutional networks. In comparison, our whole model consists of 6 convolutional networks. Even without a complex comparison, it can be concluded that the model proposed in this work can also be used in the design of online applications. However, this research, like other research, has shortcomings. One of the limitations of this research is the limited database, if you have a large database, there will be no need to increase the data by using GAN networks. Classification of two classes and not using multi-class scenarios can be called the next limitation of this research. In the future, we plan to evaluate the relevant research in real-time applications and consider more scenarios for image classification.”

Which is highlighted in Discursion section, page 13 and lines 347-376.

  1. 19. Authors need to include a section or subsection dedicated to the limitations or constraints of their experiments and probable ways to solve those issues (the limitations they have provided are at the end part of the conclusion, which is not the place they should have been).
  • ⎫ The manuscript is revised based on this comment. Yes, the opinion of the esteemed reviewer is absolutely correct, according to the opinion of my esteemed referee, a section under the title of discussion was considered in the article, and limitations and suggestions for future work were transferred to this section. 

Which is highlighted in Discursion section, page 13 and lines 371-376.

  1. 20. Authors need to make sure they have provided citations and taken permission from all the images from other sources.  
  • ⎫ The manuscript is revised based on this comment. The databases were obtained under the permission of the Ethics Committee of Tabriz University of Medical Sciences, according to the code of ethics 1399.45.AC. Also, all sources used in the article are cited.

Round 2

Reviewer 2 Report

The authors have responded to all the queries satisfactorily and made significant revisions. To my opinion, the manuscript is in much better shape and ready to be published after an overall grammar/style check.

Author Response

Based on the opinion of the honorable reviewer, the manuscript was checked and corrected in terms of grammar and spelling. 

Best Regards
